DATA RELEASE

# 1D and 2D NMR spectra of coffee from 27 countries

Javier Osorio[1], Victoria A. Arana[2,†], Jessica M. Medina[2,†], Rodrigo Alarcon[3], Edgar Moreno[3] and Julien Wist[1,4,*]

1 Chemistry Department, Universidad del Valle, 760008 Cali, Colombia
2 Chemistry Program, Basic Science Faculty, Universidad del Atlantico, 081007, Puerto Colombia, Colombia
3 Almacafé S.A., Bogotá D.C., Colombia
4 Australian National Phenome Centre, Computational and Systems Medicine, Health Futures Institute, Murdoch University, Harry Perkins Building, Perth WA 6150, Australia

## ABSTRACT

Between 2012 and 2014, 715 green coffee samples were gathered by Almacafé S.A. (Bogotá, Colombia) from 27 countries. These were analysed at the nuclear magnetic resonance (NMR) laboratory at Universidad del Valle (Cali, Colombia). Over 1000 methanolic coffee extracts were prepared and 4563 spectra were acquired in a fully automatic manner using a 400 MHz NMR spectrometer (Bruker Biospin, Germany). The dataset spans the variance that could be expected for an industrial application of origin monitoring, including samples from different harvest times, collected over several years, and processed by at least two distinct operators. The resulting 1D and 2D spectra can be used to develop and evaluate feature extraction methods, multivariate algorithms, and automation monitoring techniques. They can also be used as datasets for teaching, or as a reference for new studies of similar samples and approaches.

**Subjects** Metabolomics and Proteomics, Biochemistry, Cheminformatics

## DATA DESCRIPTION

### Context

In 2005, the Colombian Coffee Federation (Federación Nacional de Cafeteros [FNC], in Spanish) applied for a Protected Geographical Indication (PGI). This was an additional move in a branding strategy, initiated decades ago, with the registration of well-known brands such as "Juan Valdez" and "100% Colombian coffee". The PGI was granted in 2007 and the strategy was successful, as evidenced by the growing popularity of Colombian coffees around the world. All the while most coffee was sold as blends, monitoring the origin of beans was not a concern. However, during the last two decades, there has been a growing interest in specialty coffees, with businesses converting the origin into a major sales argument. There is therefore a requirement for screening methods to trace the origins of coffee beans.

In 2010, The Logistic and Quality Control Office of FNC, Almacafé S.A., began screening analytical platforms for this purpose. At that time, near infrared spectroscopy was the best-known approach, followed by conventional hyphenated techniques, such as gas or high-performance liquid chromatography coupled to mass spectrometry, and by isotope ratio mass spectrometry (see Table 1 in [1]). Little was known about nuclear magnetic resonance (NMR). Two studies reported promising results, one by Consonni *et al.* [2] using

**Submitted:** 26 October 2020

\* Corresponding author. E-mail: julien.wist@correounivalle.edu.co

† Contributed equally.

Preprint submitted at https://doi.org/10.21203/rs.3.rs-1146879/v1

**Figure 1.** Geographical areas and sample distribution covered or partially covered by this dataset.

[1]H-NMR and another by Wei and coworkers [3] using [13]C-NMR, but with small data sets (40 and 60, respectively).

In the following years, Almacafé gathered 715 samples from 27 countries (312), and from most coffee-producing areas in Colombia (403) as represented in Figure 1. At the same time, our laboratory tested and evaluated different protocols for sample preparation and data acquisition. The aim of this collaboration was to evaluate NMR as an analytical platform for large scale monitoring of the origin of coffee samples. The major outcomes can be found in Arana *et al.* [4].

Later, other techniques were benchmarked to find cheaper alternatives to NMR that would produce comparable results. Mid-infrared attenuated total reflectance (mIR-ATR) spectroscopy showed promise, with the ability to be portable in future [1, 5]. In acquiring most of the data, low-field NMR was deemed suitable for a long-term study; however; it is a promising technique [6] that should be considered for future research.

## The dataset

The dataset presented here was acquired between 2012 and 2014 at the Nuclear Magnetic Resonance (NMR) laboratory at Universidad del Valle (Cali, Colombia). Green coffee samples were collected from 27 countries (see Figure 1, Tables 1 and 2) by Almacafé S.A. (Bogotá, Colombia), during the same period. The total dataset comprises 4563 original folders (with the original hierarchy determined by Bruker Biospin), two files (tsv format), with descriptions of samples and spectra, and two datasets (tsv format) of 48 and 89 spectra as examples.

Several batches of samples were prepared (see Table 3). Within a batch, all coffee samples were prepared and spectra were recorded following the same protocol. Four types of nuclear magnetic resonance (NMR) experiment were performed, as described in the Methods and shown in Table 4. Since the parameter set enforces all other experimental parameters, it is sufficient to ensure that spectra are compared with same parameter set (the abbreviation "meoh" in filenames always refers to coffee samples extracted in methanol). The resulting spectra for each sample, including replicates, were stored in a folder (see Figure 2 and below).

**Table 1.** Distribution of the 715 samples by countries and species.

| Country | Species | Counts |
|---|---|---|
| Brazil | Unknown | 7 |
|  | Arabica | 24 |
|  | Robusta | 4 |
| Cameroon | Robusta | 1 |
| China | Arabica | 5 |
|  | Liberica | 1 |
|  | Robusta | 1 |
| Colombia | Unknown | 7 |
|  | Arabica | 388 |
|  | Decaf. Arabica | 2 |
|  | Catimor | 4 |
|  | Hibrido de Timor | 2 |
| Côte d'Ivoire | Robusta | 1 |
| Costa Rica | Arabica | 8 |
| Dominican Republic | Arabica | 7 |
|  | Robusta | 6 |
| Ecuador | Unknown | 3 |
|  | Arabica | 4 |
|  | Robusta | 1 |
| El Salvador | Arabica | 4 |
| Ethiopia | Arabica | 3 |
| Guatemala | Unknown | 3 |
|  | Arabica | 20 |
|  | Robusta | 2 |
| Hawaii (USA) | Arabica | 2 |
| Honduras | Unknown | 7 |
|  | Arabica | 10 |
| India | Robusta | 6 |
| Indonesia | Unknown | 1 |
|  | Arabica | 2 |
|  | Robusta | 4 |
| Jamaica | Arabica | 2 |
| Kenya | Arabica | 1 |
| Mexico | Arabica | 6 |
|  | Robusta | 2 |
| Nicaragua | Unknown | 1 |
|  | Arabica | 3 |
| Panama | Arabica | 11 |
| Peru | Unknown | 3 |
|  | Arabica | 17 |
| Unknown | Unknown | 64 |
|  | Arabica | 8 |
|  | Mixture | 18 |
|  | Robusta | 5 |
|  | Hibrido de Timor | 1 |
| Tanzania | Unknown | 2 |
|  | Arabica | 1 |
|  | Robusta | 1 |
| Togo | Robusta | 1 |
| Uganda | Arabica | 1 |
|  | Robusta | 4 |
| Venezuela, RB | Unknown | 3 |
|  | Arabica | 1 |
| Vietnam | Robusta | 17 |
|  | Decaf. Robusta | 1 |
| Zambia | Arabica | 1 |

**Table 2.** Distribution of the 715 samples by species and varieties.

| Batch | Variety | Counts(1D NOESY) |
|---|---|---|
| Unknown | Unknown | 99 |
| | EDERWISE | 1 |
| | SNOW BERRY | 1 |
| Arabica | Unknown | 393 |
| | BOURBON | 6 |
| | CASTILLO | 38 |
| | CATUAI | 2 |
| | CATURRA | 25 |
| | CATURRA-COLOMBIA | 1 |
| | CATURRA-COLOMBIA-CASTILLO | 1 |
| | CATURRA(30)-TIPICA(70) | 1 |
| | CATURRA(40)-TIPICA(20)-OTRA(40) | 1 |
| | CATURRA(70)-CASTILLO(30) | 1 |
| | CATURRA(70)-TIPICA(15)-CASTILLO(15) | 1 |
| | CATURRA(80)-CASTILLO(20) | 5 |
| | CATURRA(90%)-CASTILLO(10%) | 1 |
| | COLOMBIA | 2 |
| | COLOMBIA (30)-CATURRA (70) | 1 |
| | COLOMBIA(3)-CATURRA(90)-OTRA(7) | 1 |
| | COLOMBIA(30)-CATURRA(70) | 1 |
| | COLOMBIA(30)-TIPICA(50)-BOURBON(20) | 1 |
| | COLOMBIA(50)-CASTILLO(50) | 1 |
| | COLOMBIA(50)-CATURRA(50) | 1 |
| | COLOMBIA(59 %)-CATURRA(41%) | 1 |
| | COLOMBIA(65)-CASTILLO(35) | 1 |
| | COLOMBIA(70)-CATURRA(30) | 1 |
| | COSTA RICA 95 | 11 |
| | CRUCE | 10 |
| | ETIOPE | 1 |
| | GEISHA | 10 |
| | LAURINA | 3 |
| | MANDHELING | 1 |
| | MARAGOGIPE | 2 |
| | MOGIANA | 1 |
| | Ebano Verde | 3 |
| Decaf. Arabica | Unknown | 2 |
| Liberica | Unknown | 1 |
| Mixture | Unknown | 18 |
| Robusta | Unknown | 51 |
| | CONILON | 5 |
| Decaf. Robusta | Unknown | 1 |
| Catimor | Unknown | 4 |
| Hibrido de Timor | Unknown | 3 |

## Folder names and structure

Folder names abide by the following convention:

(1) a six-character code, comprising two capital letters followed by four digits representing a unique identifier for each sample. The samples received (715) were named as ACXXXX for Almacafé, while the samples prepared in Cali (mixtures as described in the Adulteration section of the Methods) were named as UVXXXX after Universidad del Valle.

(2) a section containing information on the sample, depending on the batch.

(3) a seven-digit unique identifier used by our laboratory information management system [7] to distinguish several spectra belonging to the same extract; either in different types of experiments or repetitions.

(4) a 10-character key used to import data into the database.

The four sections are separated by "_". The second section may contain more than one field, also separated by "_". To unequivocally identify the data, only the first and the penultimate sections are relevant.



**Table 3.** Number of spectra recorded in each batch.

| Batch | Counts(all) | Counts(1D NOESY) | Counts(J-RES) |
|---|---|---|---|
| Cafestol | 3 | 1 | 0 |
| CURVE-roasted | 111 | 30 | 21 |
| Mix_arabica_meoh_Roasted | 64 | 16 | 16 |
| Mix_meoh_Roasted | 584 | 146 | 146 |
| Origin_meoh_green | 435 | 110 | 105 |
| Origin_meoh_Roasted | 1841 | 461 | 455 |
| ORIGIN-MeOH | 530 | 134 | 128 |
| Quantification-green-MeOH | 40 | 10 | 10 |
| Quantification-roasted-MeOH | 64 | 16 | 16 |
| Quantification-roasted-MeOH-2 | 279 | 70 | 69 |
| Referencia-Junio-2-2012 | 3 | 1 | 0 |
| Reproducibility_2013 | 24 | 6 | 6 |
| Reproducibility-green-120713 | 19 | 6 | 1 |
| Reproducibility-green-120801 | 6 | 2 | 1 |
| Reproducibility-roasted-120713 | 37 | 12 | 0 |
| Reproducibility-roasted-120801 | 15 | 5 | 0 |
| Reproducibility-roasted-120808 | 9 | 3 | 0 |
| Reproducibility-roasted-MeOH | 18 | 6 | 0 |
| Roasted-MeOH | 64 | 16 | 16 |
| roastingEffect | 417 | 105 | 104 |

**Table 4.** Number of spectra recorded grouped by pulse sequence.

| Name of experiment | Name of parameter set | Pulse sequence file name | Dimension | Counts |
|---|---|---|---|---|
| J-Resolve [8] | COFFEE_meoh_jres4.lims | jresgppsqf.2 | 2D | 1094 |
| 1D NOESY[9] | COFFEE _meoh_noesyp64.lims | noesygpps1d.comp | 1D | 1156 |
| 1H-NMR | COFFEE_meoh_pilot1 | zg30 | 1D | 1157 |
| 1H-NMR with pre-saturation [10] | COFFEE_meoh_pilot2 | zgpsd0 | 1D | 1156 |

Bruker files are read by several packages [11–13] and software. Reading and displaying processed Bruker spectra is readily achieved by reading two files. The typical folder structure is shown in Figure 2.

Each folder contains a subfolder for each different experiment performed (by Bruker convention, these folders and subfolders are referred to as "expname" and "expno". Each subfolder contains files and one folder. "acqus" contains information about the experimental conditions for acquisition and "fid" (or "ser" for 2D analysis) contains the actual free induction decay (FID), i.e., the raw data. The data after processing, Fourier transform, phase-correction and baseline correction, are stored in the "pdata/1/" subfolder.

### Extracting spectra

Five parameters in the file "pdata/1/procs/" are needed to read and display the processed spectrum stored in file "pdata/1/1r" for 1D data ("pdata/1/2rr" for 2D). Figure 3 shows two typical spectra, from Arabica coffee in black and Robusta coffee in green.

(1) BYTORDP: endianness, 0 for little and 1 for big
(2) FTSIZE: the length of the data buffer
(3) NC_proc: the scaling factor. A value of 3 indicates that the data have been halved three times, while a negative value indicates that data have been multiplied by 2n times.

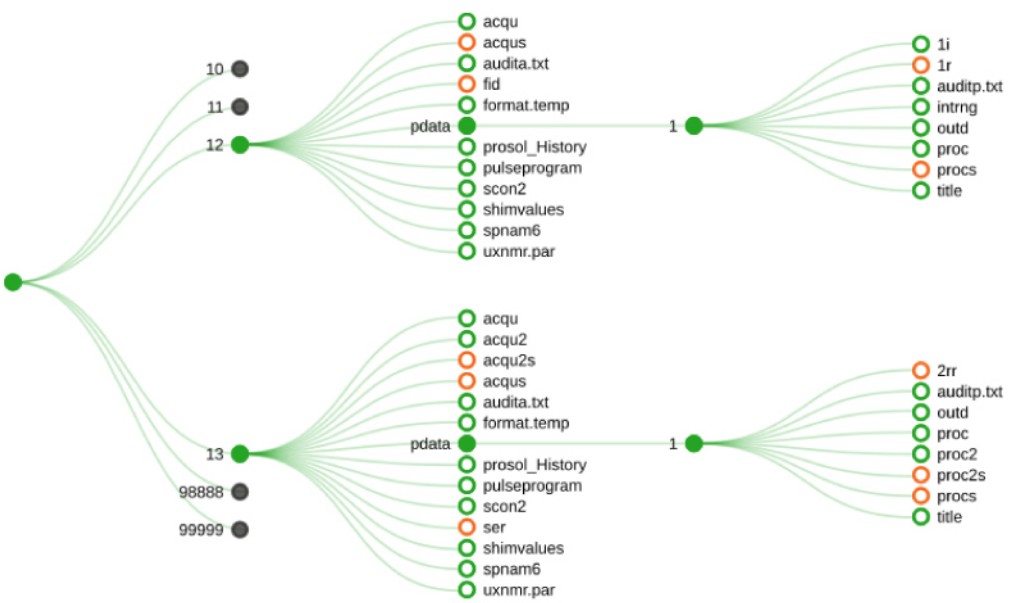

**Figure 2.** Structure of each folder. Filled circle represents folder, while open circles are for files. Folders with grey circles have similar structure (not represented here for clarity).

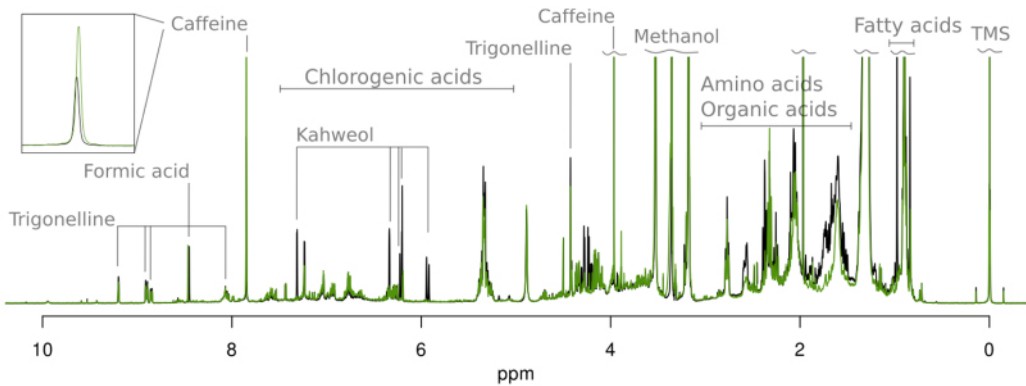

**Figure 3.** Two typical spectra of Arabica (black) and Robusta (green) coffee extracts in non-deuterated methanol. Signals assigned to Kahweol can be used for discrimination. In addition, caffeine content is expected to be higher in Robusta species, as observed in the insert.

(4)  iv. SF: the carrier frequency [MHz]

(5)  SW_p: the spectral width (or the observation window) [Hz]

(6)  OFFSET: the offset of the origin [ppm].

The *x* axis in ppm ranges between OFFSET and OFFSET–SWp/SF with increments of (SWp/SF)/(FTSIZE–1). In addition, useful information about the NMR experimental conditions can be extracted from the acqus file contained in each folder.

- ##$EXP = ⟨X⟩ is the name of the parameter set determined by ICON NMR (the Bruker automation software; see Table 4)



- ##$PULPROG = ⬚X⬚ contains the name of the pulse sequence applied to record the spectra (see Table 4)
- •##$RG = X describes the value of the receiver gain
- ##$SOLVENT = ⬚X⬚ is the name solvent parameter set used to lock the frequency and thus gives information about the solvent used for extraction.

## Samples description file (metadata)

The file "curatedCodes.tsv" contains all the available information (715 rows and 53 columns ) about the samples received from Almacafé. Only the comments column has been removed to meet privacy concerns. The most important fields, *country*, *department*, *city*, *species*, *variety* and *dateReception* were homogenized and curated; the others were left with original values.

## Experiment list

To facilitate re-use of the data, a file "curatedExtractedNames.tsv" (4563 rows and 23 columns ) is provided, which contains the path to each experiment (expname/expno) aligned with basic information about the sample, such as: *ID, entryID, expNo, parameterSet, pulseProgram, solvent, baseFrequency, frequency, temperature, spectralWidth, receiverGain, date, path, batchID, dateReception, name, country, department, species, variety, composition, roastingTemperature,* and *roastingColor*.

## Test dataset

Small datasets, "colombiaOthers48.tsv" and "robustaArabica89.tsv", are provided as examples or for quick evaluation of the usefulness of the data. The first three columns contain the path, the ID and the group. The remaining columns contain the spectra trimmed from 0.01 to 10 ppm  and after removal of the methanol region between 3.15 and 3.6 ppm . The column headers are the ppm values. Figure 4 shows 10 original spectra (without any modification) for illustration purpose. Note that this figure allows readers to interact with the data, to zoom in and out and perfrom standard operations thanks to a react component [14].

## METHODS

Several protocols and data analysis pipelines were evaluated, and several research questions were raised as our curiosity was piqued by preliminary results. Some of them were explored as reflected by the inhomogeneous granularity of the metadata ("curatedCodes.tsv") and the large number of reported batches (see Table 3).

## Sample recollection

Coffee samples (*Coffea arabica*, NCBI:txid13443 and *Coffea canephora* var. Robusta, NCBI:txid308126) were recollected by the Laboratory for Quality Control at Almacafé according to availability. This laboratory monitors the quality of coffee beans during processes that take place prior to exportation. As sole contractor of the FNC, Almacafé must monitor all the coffee harvested and exported by the FNC, and therefore has access to coffee samples from all regions of Colombia. Furthermore, as major producer and worldwide exporter, the FNC and Almacafé have access to international partners and it was thus able to gather samples from other producing countries.

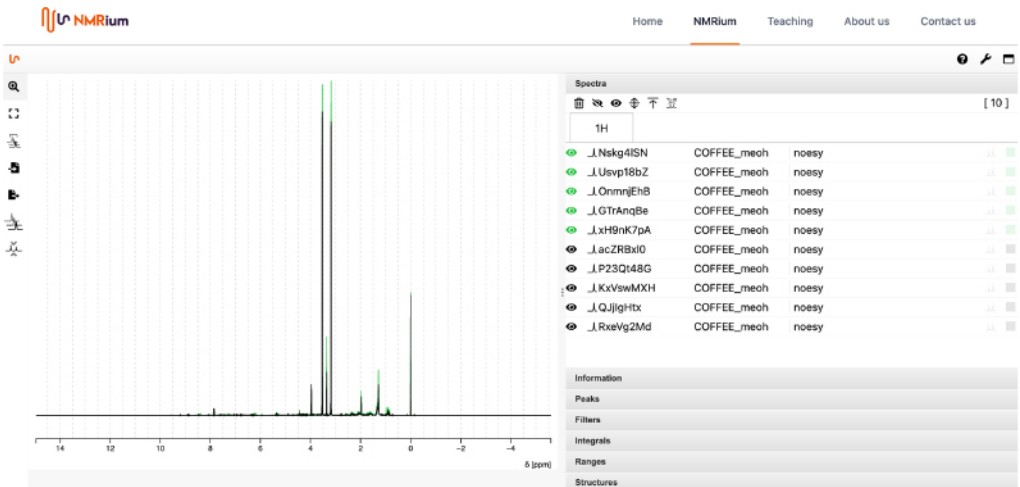

**Figure 4.** 89 original spectra (no regions removed) as used to produce the "robustaArabuca89" test set available to browse in the nmrium tool. Click in the figure to zoom in and out or to select/unselect a spectrum. https://www.nmrium.org/nmrium#?nmrium=https://dl.dropboxusercontent.com/s/2j4hrqqzuj08og0/arabica-robusta-small.nmrium?dl=0

## Sample preparation for NMR

Obtaining the fastest, most robust, and cheapest protocol to prepare samples was challenging. This included performing solid–liquid extraction at ambient temperature (to avoid controlling an additional parameter) and in non-deuterated solvent (to lower the cost per experiment). Best results were obtained using centrifugation to remove solid particles [4] after extraction. Filtration, considered initially, was discarded owing to poor reproducibility (see Table 3, Reproducibility-*).

Different solvents were compared for extraction: water, chloroform and methanol. Methanol extracts contained the greatest number of signals, thus methanol was chosen as the sole solvent for the rest of the project. However, chloroform produced interesting spectra, extracting apolar compounds, such as fatty acids, which were later found to be important for origin determination.

Both green and roasted beans were considered. Ideally, green coffee samples (Table 3: Origin_meoh_green) would be preferred to avoid the roasting process. However, this process was already well implemented and extensively used by Almacafé, plus the ball grinders required to grind green beans were not widely available at the time. Since green coffee extracts were showing rapid oxidation while waiting for acquisition, and – more importantly – samples to be analysed from the market are most likely to already be roasted, our samples were roasted according to the standard procedure for cupping and ground to a very fine powder (Table 3: *Origin_meoh_Roasted*).

In addition to regular solution-state $^1$H-NMR, other techniques were evaluated, such as solid-state magic angle spinning (MAS), and high-resolution MAS (HR-MAS). A benefit of these two techniques is that solid–liquid extraction is not needed, thus reducing the risk of contamination, and saving time and money. However, since these spectra were recorded on different machines (kindly made available by Bruker Biospin), they are not are not listed here.



### Data acquisition

NMR allows us to observe nuclear spins, particularly those of protons. Traditionally, chemists have dissolved the substance of interest in deuterated solvents, i.e., solvents in which all protons have been replaced by deuterons, which are invisible to NMR. This means only the molecules of interest contribute to the signal and thus to the spectra. Since the solvent is always much more concentrated than the substrate, its signal will obfuscate the signal of interest, in the same way as the sun makes stars invisible during the day. Deuterated solvents are much more expensive and thus less suitable for large-scale screening, especially in developing economies. Therefore, to obtain this dataset, extracts were prepared using regular methanol (chromatographic grade), while spectra were acquired using a solvent signal suppression scheme [10, 15, 16].

To achieve robust and completely automatic (without human supervision) suppression of the solvent signal, three experiments were performed for each sample: (*COFFEE_meoh_pilot1*) to estimate the resonance frequency of the solvent, (*COFFEE_meoh_pilot2*) to determine this frequency with greater accuracy by applying light suppression, and (*COFFEE_meoh_noesyp64.lims*) to acquire the spectra by suppressing the solvent at that frequency [9]. For some samples, an additional 2D experiment was performed: *COFFEE_meoh_jres4.lims*. 2D experiments encode information about chemical shifts and scalar couplings in two dimensions, to highlight correlations and reduce peak overlap. J-resolved experiments provide a wealth of information about scalar coupling constants [8] and have been less exploited owing to their size and because peak-picking and peak-fitting are more complex to perform in 2D.

An important technical note: during data acquisition, the spectrometer experienced a technical issue that modified the carrier frequency for the subsequent experiments. As such, data for some samples were re-acquired for the sake of publication. This also provided a unique opportunity to assess how carrier frequency can affect the field stability of the final models. Information about the carrier frequency can be found for each individual spectrum, in its parameter file "acqus" under the name "SFO1" (see Section "Extracting spectra"). Before the incident, the frequency was 400.13188235 MHz, and it stabilised to 400.02188249412 MHz, after the ice was removed and the pressure in the Dewar returned to normal. For simplicity, this information has been included in the file "curatedExtractedNames.tsv" in the column *baseFrequency*.

Full details about sample preparation and data acquisition can be found elsewhere [4] and within the original files, in particular in the "acqus" files available for each experiment.

### Determination of coffee origin

The focus of Almacafé is to determine the origins of coffee. This represents the core of this study, and thus the largest share of the data, with more than 2000 acquired spectra (Table 3: Origin_meoh_green, Origin_meoh_Roasted and ORIGIN_MeOH). Considerable effort was dedicated to covering a vast geographical area. Special care was taken to include samples from neighboring areas that are expected to share most similarities. When available, information about post-harvest processing, such as washing and drying, was recorded, as it is expected to strongly affect the final product. In addition, the numerous samples available from Colombia allows one to look into origin by department or by region with similar agro-climatological characteristics. For a quick look into this batch, a test dataset colombiaOthers48.tsv was prepared by randomly picking 24 samples from Colombia and 24 from other countries.

### Adulteration

A second interest of Almacafé's Laboratory for Quality Control is the possible adulteration of Colombian Arabica coffee with lower quality Robusta (*Coffea canephora* var. Robusta, NCBI:txid308126) beans imported to Colombia for internal consumption. Considerable literature has been dedicated to detecting such adulteration [17], and it was deemed to be a natural starting point for comparison with our results. Therefore, 252 samples (Table 3: Mix_arabica_meoh_Roasted, Mix_meoh_Roasted, Quantification-roasted-MeOH and Quantification-roasted-MeOH-2) were prepared by adulterating Arabica coffee with concentrations of Robusta ranging from 5–90%.

Since the species is expected to have a larger effect size than the origin, a first classification may be performed to exclude all non-Arabica samples (including adulterated ones) prior to modeling the origin [4]. Cafestol is a biomarker for discriminating Robusta from Arabica [18, 19]. Included in our dataset is a reference spectrum that was prepared in similar conditions and can be used to help identify signals. For a quick look into this batch, a test dataset robustaArabica89.tsv was prepared by randomly picking 45 samples from Colombia and 45 from other countries. Note: one sample with poor shimming was removed.

### Effect of roasting

During this collaboration, the authentication of several commercial samples from retailers around the world was conducted by Almacafé. Some of these samples are listed here as Unknown. The samples were first flagged as suspicious after cupping, and were then sent for analysis. Although the classification performed well in most cases, some samples were difficult to classify (for example, several decaffeinated samples were found, which confused the cupping process).

These latter samples raised the question of how the roasting process might confuse the models. Indeed, the samples were roasted by the retailers, and not according to the standard process used to build the classifier. A test was thus conducted (see Table 4: RoastingEffect and CURVE-roasted) to estimate the impact of roasting temperature and final colour [20] on the resulting profiles (spectra). It was proven possible to contain this effect by excluding the features of the spectra most affected by the degree of roasting (see chapter 3.5 of [21]), although, beyond some color (very dark) samples could not be modelled accurately.

### Replicates

Only considering experiments from the Origin_meoh_Roasted batch and acquired with noesygpps1d.comp, more than 83 (out of 353) samples were acquired with replicates, either to monitor the long-term stability of the process and to estimate its associated experimental noise, or because the automation failed to generate poor quality spectra. All these spectra are shared here and can be used to develop and test algorithms that can detect failed acquisition and to detect batch or run effects, using the acquisition time available in the "acqus" files.

### DATA VALIDATION AND QUALITY CONTROL

The data presented here were recorded during a collaboration project between the Laboratory of Quality Control of Almacafé S.A. (Bogotá, Colombia) and the NMR Laboratory at Universidad del Valle (Cali, Colombia). Some of the outcomes of this project can be found here [1, 4, 5], along with more details about the protocols used for sample preparation and the experimental setup to acquire the data.

## REUSE POTENTIAL

To the extent of our knowledge, this dataset represents the largest available database of NMR spectra of coffee samples. As such, it could be used as a reference baseline for future studies, and to benchmark data reduction methods or classification algorithms. The size and complexity of the dataset (a large quantity of sparse metadata covering numerous degrees of freedom) makes it a useful resource for teaching. This motivated us to release the complete dataset, including spectra acquired during validation of the methodology.

## DATA AVAILABILITY

The dataset supporting the results of this article is available at Zenodo [22]. For further details, or complementary information, please contact the corresponding author.

## DECLARATIONS
## LIST OF ABBREVIATIONS

FID: free induction decay; FNC: Federación Nacional de Cafeteros (Colombian Coffee Federation); NMR: nuclear magnetic resonance; NOESY: nuclear Overhauser effect spectroscopy; PGI: Protected Geographical Indication.

## ETHICAL APPROVAL

Not applicable.

## CONSENT FOR PUBLICATION

Not applicable.

## COMPETING INTERESTS

The authors declare no competing interests. Rodrigo Alarcon and Edgar Moreno were both working for Almacafé at the time when samples were collected.

## FUNDING

This research was funded by Almacafé S.A., Bogotá, Colombia.

## AUTHORS' CONTRIBUTIONS

Conceptualization: JW, RA, EM; data acquisition: VA and JM; data curation: JW and JO; project funding and supervision: JW; data analysis: VA, JM, JO and JW; drafting the manuscript: JW; revision and edition: JO and VA.

## ACKNOWLEDGEMENTS

Not applicable.

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
