## [Reviewer Report]

Reviewer name and names of any other individual's who aided in reviewer Reza SalekDo you understand and agree to our policy of having open and named reviews, and having your review included with the published papers. (If no, please inform the editor that you cannot review this manuscript.)YesIs the language of sufficient quality?NoPlease add additional comments on language quality to clarify if needed
There are several sentences in the manuscript that can benefit from a professional edit. Are all data available and do they match the descriptions in the paper? YesAdditional CommentsIdeally would be better to put the link for the files in the text rather than as a reference. Are the data and metadata consistent with relevant minimum information or reporting standards? See GigaDB checklists for examples <a href="http://gigadb.org/site/guide" target="_blank">http://gigadb.org/site/guide</a>YesAdditional CommentsCan the data set in Zenodo be structured differently, and placed in different folders? Currently, it isn't easy to find the data and associated metadata. Is the data acquisition clear, complete and methodologically sound?YesAdditional CommentsIs there sufficient detail in the methods and data-processing steps to allow reproduction?NoAdditional CommentsIdeally, need more parameter details, particularly for the data acquisitions and ideally for data processing. Is there sufficient data validation and statistical analyses of data quality? YesAdditional CommentsNAIs the validation suitable for this type of data?YesAdditional CommentsNAIs there sufficient information for others to reuse this dataset or integrate it with other data?YesAdditional CommentsAny Additional Overall Comments to the AuthorThank you for sharing your datasetRecommendationMinor Revision

---

## [Reviewer Report]

Upload additional filesDRR-20201004/form/Review_letter_GigaByte.pdfReviewer name and names of any other individual's who aided in reviewer Nils SchloererDo you understand and agree to our policy of having open and named reviews, and having your review included with the published papers. (If no, please inform the editor that you cannot review this manuscript.)YesIs the language of sufficient quality?YesPlease add additional comments on language quality to clarify if needed
In some points, an explanation of technical terms may be included and there are a few typographical errors, which are both listed in the attached document Are all data available and do they match the descriptions in the paper? YesAdditional CommentsAre the data and metadata consistent with relevant minimum information or reporting standards? See GigaDB checklists for examples <a href="http://gigadb.org/site/guide" target="_blank">http://gigadb.org/site/guide</a>YesAdditional CommentsIs the data acquisition clear, complete and methodologically sound?YesAdditional CommentsIs there sufficient detail in the methods and data-processing steps to allow reproduction?YesAdditional CommentsIs there sufficient data validation and statistical analyses of data quality? YesAdditional CommentsIs the validation suitable for this type of data?YesAdditional CommentsIs there sufficient information for others to reuse this dataset or integrate it with other data?YesAdditional CommentsAny Additional Overall Comments to the AuthorThe dataset containing nuclear magnetic resonance (NMR) spectra from coffee samples, submitted by Osorio et al. is of high interest for experts working in the field of NMR spectroscopy, chemometrics and statistical analysis of (e.g.) food and agricultural products. Therefore, the material delivered with this publication will be very useful as training datasets and for education/teaching as intended by the authors. The quality of datasets is high and since the samples are delivered as complete original data, conversion into other NMR data formats can be easily achieved if needed. There are a few minor observations, all of them concerning typos, which are detailed in the attached file, but no objections concerning scientific contents.RecommendationAccept